Deep cryptic diversity in the Craugastor podiciferus Species Group (Anura: Craugastoridae) of Isthmian Central America revealed by mitochondrial and nuclear data

Arias Erick 1 2 3
Crawford Andrew J. 4 5 6
Hertz Andreas 7
Parra Olea Gabriela gparra@ib.unam.mx 3
1 Escuela de Biología, Universidad de Costa Rica , San José , Costa Rica
2 Museo de Zoología, Centro de Investigaciones en Biodiversidad y Ecología Tropical, Universidad de Costa Rica , San José , Costa Rica
3 Zoology, Instituto de Biología, UNAM , Mexico City , Mexico
4 Departamento de Ciencias Biológicas, Universidad de los Andes , Bogotá , Colombia
5 Smithsonian Tropical Research Institute , Panama City , Republic of Panama
6 Círculo Herpetológico de Panamá , Panama City , Panama
7 Department of Biology, University of Massachusetts at Boston , Boston , MA , United States of America
Morrone Juan J.
Electronic publication date: 2025 Jan 17
Publication date: 2025
Volume: 13
Electronic Location ID: e18212
Received 2023 Aug 16; Accepted 2024 Sep 11
Copyright: ©2025 Arias et al.
Copyright year: 2025
Copyright holder: Arias et al.
License: This is an open access article distributed under the terms of the Creative Commons Attribution License, which permits unrestricted use, distribution, reproduction and adaptation in any medium and for any purpose provided that it is properly attributed. For attribution, the original author(s), title, publication source (PeerJ) and either DOI or URL of the article must be cited.
License URL: https://creativecommons.org/licenses/by/4.0/

Keywords: Ancestral area reconstruction, Costa Rica, Historical biogeography, Candidate species, Species delimitation, Systematics, Taxonomy

Funding: The CONACyT (CVU/Becario) 626946/330343 and the Programa de Innovación y Capital Humano para la Competitividad PINN-MICITT PED-0339-15-2 PAPIIT-UNAM IN208024 The National Geography Society W-346-14 The CONACyT (CVU/Becario) 626946/330343 and the Programa de Innovación y Capital Humano para la Competitividad PINN-MICITT (PED-0339-15-2) supported Erick Arias. Laboratory work was funded by a grant from PAPIIT-UNAM (IN208024) to Gabriela Parra Olea. Fieldwork was supported by the National Geography Society (Grant number W-346-14). There was no additional external funding received for this study. The funders had no role in study design, data collection and analysis, decision to publish, or preparation of the manuscript.

==============================
The Craugastor podiciferus Species Group contains eleven species of terraranan frogs distributed from eastern Honduras to eastern Panama. All species have remarkable color pattern polymorphisms, which may contribute to potential taxonomic problems. We performed exhaustive sampling throughout the geographic distribution of the group to evaluate the phylogenetic relationships and biogeographic history of all named species based on two mitochondrial markers and nuclear ddRAD loci. We also implemented various species delimitation methods to test for the presence of unconfirmed candidate species within the group. Molecular phylogenetic analyses showed that the group contains four major clades. All currently named species are supported by molecular data, yet species richness within the group is clearly underestimated. Species delimitation was discordant between the mitochondrial and nuclear datasets and among analytical methods. Adopting a conservative approach, we propose that the C. podiciferus species group contains at least 12 unconfirmed candidate species. Ancestral area reconstruction showed that the group originated and diversified in the highlands of the Talamancan montane forest ecoregion of Costa Rica and western Panama.

Introduction

Species diversity is not homogeneously distributed over the globe. The American tropics have among the highest biodiversity in the world, including seven of 25 biodiversity hotspots (Myers et al., 2000). The Mesoamerica biodiversity hotspot includes the Isthmian Central America (ICA) region centered in Costa Rica and Panama and is renowned for its exceptional biodiversity and endemism (Bagley & Johnson, 2014). This region hosts more species of amphibians (Savage, 2002; AmphibiaWeb, 2024), reptiles (Savage, 2002; Solórzano, 2022), birds (Anger & Dean, 2010; Garrigues & Dean, 2014), insects (Doré et al., 2022), orchids (Bogarín et al., 2013; Crain & Fernández, 2020), and vascular plants (Davis et al., 1997) per area unit than almost any other place in the world. The high biodiversity of the ICA has been attributed to two major factors. First, the closure of the Isthmus of Panama allowed the Great American Biotic Interchange (GABI) between North America and South America, resulting in the coexistence of long-independent lineages of mammals and other organisms (Savage, 1966; Vanzolini & Heyer, 1985; Marshall, 1988; Webb, 2006; Pinto-Sánchez et al., 2012). Second, the long history of volcanic and orogenic activity in the ICA resulted in high geographic and climatic heterogeneity (Weyl, 1980; Herrara, 1985; Bagley & Johnson, 2014; Montes et al., 2015; García-Rodríguez et al., 2021) that has promoted in situ diversification and high levels of endemism associated with diverse habitats (Savage, 2002; Boza-Oviedo et al., 2012; Bogarín et al., 2013; Doré et al., 2022; Solórzano, 2022).

Given its small area and the relatively large and active multinational taxonomic communities working in the ICA, new species are continually discovered and described. However, extensive areas remain unexplored, and some particular groups, such as amphibians, still lack taxonomic resolution. Previous molecular systematic studies on direct-developing frogs (i.e., terraranans such as Craugastor, Diasporus, and Pristimantis) and direct-developing plethodontid salamanders (especially Bolitoglossa) have revealed extremely high levels of genetic diversity in the highlands and lowlands of the ICA (García-París et al., 2000; Crawford, 2003; Crawford, Bermingham & Polanía, 2007; Wiens et al., 2007; Wang, Crawford & Bermingham, 2008; Streicher, Crawford & Edwards, 2009; Batista et al., 2016; García-Rodríguez, Arias & Chaves, 2016).

The Craugastor podiciferus Species Group (Anura: Craugastoridae; (Hedges, Duellman & Heinicke, 2008)) is currently composed of eleven described species (Arias, Hertz & Parra-Olea, 2019). This group occurs from eastern Honduras to eastern Panama, covering a wide variety of habitats and ranging in elevation from sea level to 2,700 m (Savage & Emerson, 1970; Savage, 2002). Previous molecular studies on the systematics and taxonomy of the C. podiciferus Species Group support the presence of several undescribed species (Crawford & Smith, 2005; Arias, Hertz & Parra-Olea, 2019). Crawford (2003) found high genetic divergences among populations of C. stejnegerianus in the ICA lowlands on the Pacific coast. Streicher, Crawford & Edwards (2009) mentioned that the name C. podiciferus could mask a species complex formed by up to six distinct taxa and supported the existence of an undescribed species “Craugastor sp. B” related to C. podiciferus. Arias, Hertz & Parra-Olea (2019) reported that Craugastor sp. B of Streicher, Crawford & Edwards (2009) corresponds to C. blairi (Barbour, 1928). According to mitochondrial sequences and morphological data, Arias et al. (2016) showed that populations formerly considered part of C. stejnegerianus from southwestern Costa Rica and western Panama belong to a different species, C. gabbi (Arias et al., 2016).

The C. podiciferus Species Group (S.G.) represents an ideal model for studies of amphibian cryptic diversity, given its high local abundance, collectively wide geographic distribution, high genetic diversity, and high levels of polymorphism. To our knowledge, no molecular studies have extensively evaluated the phylogenetic relationships and the potential cryptic diversity of an amphibian species group restricted to ICA, including populations both from highlands and lowlands and using mitochondrial markers and an extensive nuclear dataset. Here, we use mitochondrial gene sequences and a genome-scale dataset to: (1) infer the phylogenetic relationships of the C. podiciferus Species Group, (2) determine the existence of overlooked species within the currently recognized species, and (3) identify the center of origin for the group and suggest plausible a historical framework for its diversification.

Materials & Methods

Taxon sampling

Tissue samples were collected for all eleven named species of the C. podiciferus Species Group in all countries where the group occurs: Honduras, Nicaragua, Costa Rica, and Panama (Fig. 1, Appendix 1). All of the specimens collected for this study were humanely euthanized through the use of a 20% lidocaine hydrochloride (Xylocaine) injection, and all efforts were made to minimize suffering. The specimens were then fixed in a 10% formalin solution and transferred to 70% ethanol for long-term storage. Tissue samples used for genetic analyses were preserved in 96% ethanol or in RNAlater™. Vouchers were deposited at the Museo de Zoología, Universidad de Costa Rica (UCR), the Division of Amphibians and Reptiles at the Field Museum of Natural History, Chicago, USA (FMNH), Círculo Herpetológico de Panamá (CH), and Senckenberg Research Institute and Nature Museum, Frankfurt, Germany (SMF). Museum collection acronyms follow Frost (2024), with the addition of the following three collectors’ field numbers: AJC referring to Andrew J. Crawford, AH referring to Andreas Hertz, and EAP referring to Erick Arias.

Figure 1 Geographic distribution of the Craugastor podiciferus Species Group from Honduras (top) through Nicaragua, Costa Rica, to central Panama (right side).

The mountains of southeastern Costa Rica are referred to as the Talamanca Mountain Range in the text. The purple circle correspond to the Craugastor bransfordii clade; green triangle = C. stejnegerianus clade; blue square = C. podiciferus clade; red star = C. aenigmaticus clade. The DEM used in this study was obtained online from CGIAR-CSI SRTM website: http://srtm.csi.cgiar.org.

Assigning specimens to species

As noted above, species in the C. podiciferus S.G. are challenging taxonomically because morphological differences among them are subtle, and many of the focal species have intraspecific color pattern polymorphism. We assigned names to clades based on type localities and general morphology, we compared tissued voucher specimens with type specimens to assign specimens to species. We included sequences of specimens from the type locality for all named species (except C. lauraster, available sequence is from ca. 50 km straight line from locality of one paratype), and we confirmed that those specimens agreed morphologically with type material. In addition, we included sequences of type localities for the synonymized species C. rearki and C. jota. We were able to confidently match voucher material to the following seven species based on type comparisons and type localities: C. blairi, C. bransfordii, C. persimilis, C. podiciferus, C. lauraster, C. stejnegerianus, and C. underwoodi. In addition, four species have been recently described using DNA sequences therefore in this case the comparisons are highly confident: C. aenigmaticus, C. gabbi, C. sagui, and C. zunigai. In total these assignments resulted in 11 species names being associated with our sampling either through confident type specimen-based identification or type material sequenced.

Mitochondrial data

Amplification and sequencing

We followed standard protocols for DNA extraction (Sambrook & Russell, 2006). We amplified fragments of two mitochondrial genes: the large subunit ribosomal RNA (16S) (Palumbi et al., 1991) and the 5′-end of cytochrome oxidase subunit I (COI) (Meyer, 2003) following standard protocols (Vences et al., 2005). PCR products were cleaned with ExoSap-IT (USB Corporation) and sequenced in both directions using amplification primers and BigDye termination reaction chemistry (Applied Biosystems). The cycle-sequencing products were column-purified with Sephadex G-50 (GE Healthcare) and run on an ABI 3500xL Genetic Analyzer (Applied Biosystems). Consensus sequences for each individual were constructed using SEQUENCHER 5.3 (Genes Codes Corp).

Phylogenetic analyses

We generated 16S and COI sequences for the C. podiciferus Species Group members. We used sequences of C. loki to root mtDNA trees, based on Crawford & Smith (2005). See Appendix 1 for a list of the examined materials, their localities, museum vouchers, and GenBank accession numbers. Sequences of each gene were trimmed at the 3′ and 5′ ends until a majority of operational taxonomic units (OTUs) had sequence data for a given character. The two genes were aligned independently using MUSCLE 3.7 (Edgar, 2004) with default parameters and then concatenated. We used PartitionFinder v2.1.1 software (Lanfear et al., 2017) and the Bayesian Information Criterion (BIC) to select the best partition scheme and the best model of sequence evolution for each partition. We used a single set of branch lengths across all partitions (branchlengths =linked), and the search for the best partition scheme used a heuristic search (scheme =greedy Lanfear et al., 2012). We defined, a priori, four subsets: one for 16S and three for COI (partitioned by codon position). The selected partition scheme and substitution models selected by PartitionFinder were used in the Bayesian MCMC phylogenetic inference.

We used maximum likelihood (ML) and Bayesian MCMC methods to infer phylogenetic hypotheses from the concatenated loci. All phylogenetic analyses were run on the CIPRES portal (Miller, Pfeiffer & Schwartz, 2010). We performed ML analyses using RAxML-HPC v8 (Stamatakis, 2014) with an unpartitioned GTR + GAMMA model of nucleotide substitution (the default model of RAxML) and the “–f a” option, which searches for the best-scoring tree and performs a rapid bootstrap analysis (1,000 bootstrap replicates) to estimate nodal support by resampling characters with replacement. A partitioned Bayesian MCMC phylogenetic analysis was performed using MrBayes 3.2.6 (Ronquist et al., 2012) with the previously selected partition scheme and substitution model (see above). Two separate analyses were run, each consisting of 50 million generations, sampling trees every 1,000 generations and using four chains with default heating parameters. We examined a time-series plot of the likelihood scores of the cold chain to check stationarity using Tracer 1.6 software (Rambaut et al., 2014). We discarded the first 25% of trees as burn-in and used the remaining trees to estimate the allcompat tree along with the posterior probabilities for each node and each parameter.

We used the program BEAST v1.8.3 (Drummond et al., 2012) to estimate a concatenated ultrametric phylogenetic tree (timetree) using an uncorrelated lognormal relaxed clock, a birth-death process tree prior, and the partition scheme and nucleotide substitution model selected previously. We ran the analysis for 50 million generations, sampling trees every 1,000 generations, and discarded the first 5,000 samples as burn-in when estimating a consensus tree. We preferred not to calibrate the ultrametric tree using mtDNA, time calibration is performed using a nuDNA dataset (see below).

Nuclear data

ddRADseq data collection

We generated ddRADseq data for 48 samples of the C. podiciferus Species Group plus one sample of C. rhodopis as an outgroup. We followed the protocol described by Peterson et al. (2012) and modified by Leaché et al. (2015). High-molecular-weight genomic DNA was further purified with RNase A, examined for quality on agarose gels, and quantified with a Qubit 2.0 fluorometer (Thermo Fisher Scientific). We used 1,000 ng of genomic DNA for each sample, except for three samples that had just 241–630 ng. We double-digested the genomic DNA with 20 units each of a rare 8-cutter, SbfI (restriction site 5′-CCTGCAGG-3′), and a common 4-cutter, MspI (restriction site 5′-CCGG-3′), in a single reaction with the manufacturer recommended buffer (New England Biolabs) for 2 h at 37 °C. Postdigestion fragments were purified with Serapure 1.5X and quantified with a Qubit 2.0 fluorometer before ligating barcoded Illumina adaptors onto the fragments.

The oligonucleotide sequences used for barcoding and adding Illumina indexes during library preparation were those employed in Leaché et al. (2015). The barcodes differed by at least two base pairs to reduce the chance of errors caused by inaccurate base calls subsequent to barcode assignment. Equimolar amounts of each sample were pooled in a 48-well plate format, with each pool containing up to eight unique barcoded samples. Each pool was purified with Serapure 1.5X, rehydrated in 50 µL, and quantified with a Qubit 2.0 fluorometer before size selection. The pooled libraries were size-selected (∼500 bp) on an e-gel (Invitrogen) according to the manufacturer’s instructions. The two external and internal lanes (next to the leader lane) in the e-gel were not used, while in the four available lines, only two different libraries were run, with a maximum of 500 ng per line. The size-selected libraries were quantified again with a Qubit 2.0 fluorometer and amplified using PCR (polymerase sequence reaction) with the primers designed by Leaché et al. (2015) and Phire Hot Start II polymerase (Thermo Fisher Scientific). The amplified libraries were purified with Serapure 1.5X and quantified with a Qubit 2.0 fluorometer.

The fragment size distribution and concentration of each pool were determined on an Agilent BioAnalyzer, and qPCR was performed to determine sequenceable library concentrations before multiplexing equimolar amounts of all 6 pools for sequencing on a single Illumina HiSeq 2000 lane (100 bp, single-end run) at the Vincent J. Coates Genomics Sequencing Laboratory at UC Berkeley.

ddRADseq bioinformatics

We processed raw Illumina reads with the software pipeline ipyrad v0.5.15 (Eaton, 2014), which consists of seven steps. Six of the 48 samples had <50,000 reads passing the quality filter and were excluded from further analyses, the remaining 42 samples were edited and filtered. The 6-bp restriction site overhang and the 5-bp barcode were removed. Bases with an accuracy of <99% (Phred quality score = 20) were converted to ‘N’ characters and reads with >9 Ns (∼10% Ns) were discarded. During steps 3–6, the reads from each sample were clustered using the program VSEARCH version 1.11.1 (https://github.com/torognes/vsearch). We determined the optimal value for the clustering parameter using the clustering threshold series approach described by Ilut, Nydam & Hare (2014) using similarity thresholds ranging from 0.85 to 0.98 for 19 randomly chosen samples. The optimal clustering threshold was 0.9 (Fig. S1A), which was used within and between sample clustering. Consensus sequences were then clustered across samples and aligned with MUSCLE version 3.8.31 (Edgar, 2004). Within steps 3–6, we also discarded loci that had >4 ambiguous or heterozygous sites (default ipyrad settings) or >2 haplotypes (to filter out paralogs) and used a minimum depth of coverage of six for genotype calls.

Following Nieto-Montes de Oca et al. (2017), we performed multiple replicates of the seventh step to determine the optimal value for the parameters: maximum numbers of SNPs allowed in a locus, maximum number of insertions/deletions allowed in among-sample clusters, and the maximum proportion of samples allowed to share a heterozygous site. We selected a maximum of 20 SNPs (Fig. S1B) per locus (default ipyrad settings). We permitted a maximum of eight insertions/deletions per locus for the assembly of the final dataset (default ipyrad settings; Fig. S1C). The number of loci retained increased roughly linearly (Fig. S1D) until it first plateaued at a value of 0.1 (corresponding to four samples), which we again chose for the final value. We followed Crotti et al. (2019) in setting the maximum missing data at 74% missing data per SNP. Crotti et al. (2019) found that the plateau with maximum support for their phylogenetic hypothesis was obtained by including SNPs with up to 70%–80% of missing data. With a maximum missing data of 74% we included SNPs were represented by at least 26% or 11 samples.

Phylogeny inference

We used maximum likelihood and Bayesian MCMC methods to estimate phylogenetic trees from the concatenated ddRAD loci, which contained 697,771 characters for 42 samples. We ran MrBayes and BEAST analyses using the same priors as in the mitochondrial data analyses (see above), except only one GTR+I+G model was used for analyzing the unpartitioned RADseq data. To evaluate the effect of the number of terminals in the topology, we also performed RAxML, MrBayes, and BEAST analyses for each of the three major clades within the C. podiciferus Species Group (C. bransfordii clade, C. podiciferus clade, and C. stejnegerianus clade; see Results). To perform the analyses by clade, we replicated the pipeline in ipyrad for each clade, maximizing the number of retained loci according to the same filters applied to the complete dataset (see above).

Relaxed molecular clock analysis

We used the program BEAST v1.8.3 (Drummond et al., 2012) to estimate a concatenated ultrametric phylogenetic tree (timetree) using the nuDNA dataset with the same prior as in Mitochondrial data: Phylogenetic analyses. We used this analysis to perform a time calibration for the C. podiciferus Species Group. The lack of C. podiciferus in the fossil record makes dating divergences based on molecular sequence data difficult. We used the dating for the group estimated by Streicher, Crawford & Edwards (2009) to estimate divergence times within the species group. Following Streicher, Crawford & Edwards (2009) and citations therein, we assumed that ICA emerged ∼25 Mya and used a secondary calibration of 20 Mya for the crown age of the C. podiciferus Species Group using a normal prior distribution with a mean of 20 Mya and standard deviation (SD) of 2 Mya to place 95% of the prior distribution on 16.7–23.3 Mya.

Species delimitation

We used six species delimitation methods to investigate species boundaries in the C. podiciferus Species Group and evaluated the effect of the phylogeny assumed. We performed analyses separately but identically on the mitochondrial DNA (mtDNA) and nuclear DNA (nuDNA) datasets. Using mtDNA, 24 combinations were performed to combine three tree inputs (BEAST, MrBayes, and RAxML) and six methods (GMYC, PTP, mPTP, BPP, ABGD, and GD (see details below)); some methods (i.e., GMYC, mPTP, BPP) have 2–4 variants which were evaluated, some methods (i.e., GMYC and BPP) only can be used with ultrametric tree (BEAST), and finally, some methods (i.e., ABGD and GD) do not use a guide tree. Using nuDNA, 37 combinations were performed to combine three tree inputs (BEAST, MrBayes, and RAxML) and four methods (GMYC, PTP, mPTP, and BPP (see details below)); some methods (i.e., GMYC, mPTP, BPP) have 2–4 variants which were evaluated, and some methods (i.e., GMYC and BPP) only can be used with ultrametric tree (BEAST). In addition, to the nuDNA dataset all the combinations of species delimitations were repeated to evaluate separately the total phylogeny and the three partial phylogenies (independent analyses of each of the three main ingroup clades—the C. bransfordii, C. podiciferus, and C. stejnegerianus clades). Except for the BPP analysis, which only were used the partial phylogenies.

We used three tree-based species delimitation methods, GMYC, PTP, and mPTP. The GMYC method (Pons et al., 2006; Fujisawa & Barraclough, 2013) infers the transition point between intraspecific (coalescent process) and interspecific (Yule process) branching rates on a time-calibrated tree. We ran the GMYC analyses separately on each of the two datasets, concatenated mitochondrial and concatenated ddRADseq data, using the web server (http://species.h-its.org/gmyc/) under the single threshold and multiple threshold GMYC models assuming the respective timetree from the BEAST analyses.

The PTP method (Zhang et al., 2013) uses the number of substitutions to identify significant changes in the rate of branching in a phylogenetic tree (which may or may not be ultrametric). We ran PTP in the web server (http://species.h-its.org/ptp/) for 500,000 generations, with thinning = 100 and burn-in = 10%. Following a conservative approach, we considered candidate species those clades with a posterior delimitation probability less than 0.01 (mitochondrial) or 0.05 (nuclear), where the posterior probability indicates that the clade in question forms a single species. Given that PTP uses any completely bifurcating tree, we used the RAxML, MrBayes, and BEAST trees to evaluate the effect of the input tree. As with the GMYC method, we performed PTP separately on the mitochondrial DNA (mtDNA) and nuclear DNA (nuDNA) datasets.

The third method, mPTP (Kapli et al., 2016), is similar to PTP but incorporates different rates of coalescence within clades, allowing different levels of intraspecific genetic diversity. Similar to PTP, mPTP runs both ML and MCMC analyses. MCMC analyses were run for 100 million generations, sampling once every 10,000 generations, and the first 2 million generations were discarded as burn-in. All ML and Bayesian analyses were run as both single and multiple rates of coalescence among species and resolved any polytomies in the input tree randomly by adding a branch of length 0.0001. For the MCMC analyses, we conservatively considered clades with a delimitation support value greater than 0.99 (mitochondrial) or 0.95 (nuclear) as species. As with the PTP analysis, we used our RAxML, MrBayes, and BEAST (consensus) trees to evaluate the effect of the assumed tree; all analyses were performed including the outgroup.

We used a fourth method, automatic barcode gap discovery (ABGD; Puillandre et al., 2012). Unlike the three tree-based methods, ABGD uses genetic distances estimated from an alignment of DNA sequences, making it difficult to apply meaningfully to our ddRAD dataset. This method was therefore used only for the mitochondrial dataset, and analyses were performed separately for 16S and COI markers. ABGD evaluates many possible ‘barcode gap’ locations that could separate intraspecific from interspecific distances. We used Pmin (0.01), Pmax (0.1), JC69 corrected distances, and a relative gap width of 1.5 (default).

For our fifth method of species delimitation, we used BPP version 3.1 (Yang, 2015; Yang & Rannala, 2010; Yang & Rannala, 2014) to jointly perform species delimitation and species tree inference under the multispecies coalescent model. We used method A10, which evaluates species delimitation from a guide tree, using a rjMCMC algorithm (Rannala & Yang, 2013). Each individual is assigned to a putative species, and the rjMCMC algorithm evaluates subtrees generated by collapsing or splitting nodes on the guide tree without performing any type of branch swapping. We used the trees generated with BEAST from the mtDNA and nuDNA data as a user-specified guide tree; mtDNA and nuDNA datasets were independently analyzed. The analysis was run for 500,000 generations (sampling interval of 5) with a burn-in period of 1000 generations. We evaluated the influence of the ancestral population size (θ) and root age (τ0) considering three different combinations of parameter values, as in Leaché & Fujita (2010). The first combination of prior distributions is θ ∼ gamma prior G (1, 10) and τ0 ∼ G (1, 10). The second combination of priors is θ ∼ G (2, 2000) and τ0 ∼ G (2, 2000). The third combination is a mixture of priors: θ ∼ G (1, 10) and τ0 ∼ G (2, 2000). In the BPP analyses of the mitochondrial data, we used algorithm 0 with the fine-tuning parameter ɛ = 15 each with 500,000 generations (saving each fifth sample) and a burn-in of 10,000 generations. With the nuclear dataset (BEAST tree), the number of generations was 100,000, and the other parameters were the same as above. Each analysis was run at least twice to confirm consistency between runs. To be conservative, only divergence events supported by posterior probabilities ≥ 0.99 for all three combinations of priors were considered for species delimitation.

As our sixth and final approach to species delimitation, we used simple genetic distances based on each mtDNA dataset separately. Although not considered a formal species delimitation algorithm, genetic distance has been used as an indicator of candidate species. For amphibians, the 16S gene fragment has been suggested as a DNA barcode marker for diversity inventories (Vences et al., 2005) to complement the newer standardized marker COI-5′ used for animals (Smith, Poyarkov Jr & Hebert, 2008). Fouquet et al. (2007) suggested a threshold of 3% in the 16S marker to identify candidate species. Vences et al. (2005) suggested a threshold of 10% in the COI barcode for identifying candidate species. Genetic distances (uncorrected p-distances, Table 1) were computed using MEGA7 (Tamura et al., 2013) for each gene separately.

Table 1 Mean uncorrected genetic distances among lineages within the Craugastor podiciferus Species Group based on mitochondrial genes 16S (above the diagonal) and COI (below the diagonal).

Genetic distances within the three major clades are highlighted in blue, purple, and green along the diagonal.

ID	16S/COI	1	2	3	4	5	6	7	8	9	10	11	12	13	14	15	16	17	18	19	20	21	22	23	
1	Craugastor aenigmaticus	—-	17.5	13.3	15.6	15.0	14.2	14.4	15.1	14.8	14.8	17.3	—-	14.2	16.7	16.3	16.2	16.8	18.7	20.3	19.6	18.9	19.6	21.7	
2	Craugastor sagui	21.8	—-	7.9	7.4	7.3	7.1	7.4	9.5	8.2	8.4	9.1	—-	11.1	17.5	12.0	11.1	12.1	12.8	16.1	16.2	16.2	16.3	18.0	
3	Craugastor zunigai	25.0	17.3	—-	3.4	7.2	6.2	7.3	9.3	9.4	9.6	8.9	—-	12.3	17.0	13.2	13.7	13.1	13.8	16.3	16.4	15.7	16.4	19.4	
4	Craugastor blairi	20.2	20.3	18.3	—-	7.6	7.4	7.6	10.5	8.3	9.9	9.4	—-	13.3	17.7	12.0	12.5	13.0	14.1	17.0	17.1	17.0	17.2	21.0	
5	Craugastor sp. Monte Verde	19.5	16.6	18.7	19.9	—-	2.5	2.0	3.7	6.6	4.4	4.5	—-	9.1	14.9	9.9	11.1	10.4	11.5	14.6	14.7	14.6	14.8	16.6	
6	C. podiciferus	22.2	17.8	18.5	19.8	12.0	—-	2.7	4.1	6.6	4.6	5.1	—-	7.6	13.6	10.1	11.2	9.6	11.5	13.2	13.2	13.1	13.2	15.9	
7	Craugastor sp. Pico Blanco	20.8	18.4	20.7	17.2	11.3	10.0	—-	2.6	6.9	4.7	3.1	—-	9.8	14.5	10.4	11.7	9.9	11.8	15.2	15.3	15.4	15.3	14.9	
8	Craugastor sp. San Gerardo	21.9	18.4	21.4	20.5	12.8	12.4	7.3	—-	7.8	5.8	4.5	—-	9.8	14.2	10.9	11.7	9.2	13.4	15.3	15.4	15.5	15.4	13.7	
9	Craugastor sp. Siola	24.0	19.8	22.8	19.6	12.8	12.9	11.4	13.7	—-	8.0	9.0	—-	11.0	15.8	9.2	10.3	10.5	11.8	13.0	11.5	11.5	11.6	16.1	
10	Craugastor sp. Chumacera	19.3	19.3	21.3	18.3	12.4	11.8	9.9	12.8	11.6	—-	7.4	—-	11.2	17.6	11.8	12.1	12.7	14.2	17.8	16.8	16.8	16.9	19.9	
11	Craugastor sp. Fila Costeña	21.2	17.8	23.4	19.9	12.5	12.3	12.0	14.4	13.8	9.6	—-	—-	12.0	16.5	11.4	12.7	10.9	12.7	14.2	15.9	15.8	16.0	16.0	
12	Craugastor sp. Quebradas	25.4	19.2	22.0	24.5	24.2	22.6	24.2	25.3	26.3	22.8	25.2	—-	—-	—-	—-	—-	—-	—-	—-	—-	—-	—-	—-	
13	Craugastor sp. Vereh	26.9	20.2	23.4	23.9	19.7	22.6	23.8	23.7	26.0	20.8	22.6	16.2	—-	5.4	6.0	6.8	5.0	9.5	11.6	10.2	10.2	10.3	10.2	
14	C. bransfordii	25.3	18.8	21.1	23.0	21.1	19.9	20.6	22.3	24.0	20.8	21.2	16.7	11.7	—-	10.4	11.3	9.4	14.4	14.1	14.1	14.1	14.2	13.4	
15	C. underwoodi	23.5	19.1	21.8	19.9	19.4	19.7	19.7	22.4	22.9	20.9	21.2	15.3	14.5	13.2	—-	4.1	5.4	8.5	11.5	10.9	10.9	11.0	12.1	
16	Craugastor sp. Fila Carbón	22.9	20.7	23.4	22.9	22.4	23.3	23.9	26.3	24.9	22.5	23.0	17.3	14.0	15.6	9.5	—-	4.3	7.7	12.2	11.6	11.5	11.6	11.6	
17	Craugastor sp. Panama	21.3	18.2	22.5	22.2	21.1	21.1	19.8	23.4	24.1	21.5	22.1	17.3	14.3	13.2	11.9	14.0	—-	7.6	11.8	10.5	10.4	10.5	8.8	
18	C. persimilis	24.2	22.4	26.5	24.0	23.0	23.8	24.9	27.4	24.1	26.2	26.4	23.8	20.8	21.8	19.1	19.6	20.9	—-	7.6	6.3	7.0	6.5	9.0	
19	C. gabbi	25.4	26.7	27.6	24.3	24.1	23.5	24.7	26.3	26.7	26.7	26.9	22.0	20.1	23.3	20.6	19.6	21.5	16.7	—-	4.2	4.3	4.3	8.0	
20	Craugastor sp. Neilly	24.9	27.5	27.4	22.6	24.6	27.2	26.9	28.9	29.3	29.8	29.2	24.2	22.4	22.9	20.5	20.4	22.6	16.3	14.1	—-	1.1	0.2	6.0	
21	Craugastor sp. Quepos	23.0	27.3	27.4	22.6	24.6	25.3	26.0	26.2	27.3	28.4	26.2	23.6	23.4	22.7	19.1	20.1	22.2	15.3	15.1	6.9	—-	1.1	6.1	
22	C. stejnegerianus	21.8	27.5	25.2	22.3	25.1	25.5	25.8	27.0	28.7	27.8	27.2	22.6	21.6	21.6	20.9	20.3	21.9	16.2	14.7	5.6	5.5	—-	6.0	
23	C. rearki	22.6	25.4	24.9	23.8	23.6	28.0	27.6	28.6	28.3	27.9	27.7	23.4	20.8	22.0	22.2	20.0	21.1	17.2	15.7	11.9	11.6	13.8	—-	

Because different species delimitation algorithms make distinct assumptions and focus on contrasting aspects of the data (e.g., distance-based vs. tree-based), and exhibit contrasting levels of statistical power (Jacobs et al., 2018), we applied multiple methods to our data sets. We should interpret these results using our expertise on the group and taking into account other data such as their morphology and ecology.

To propose a more accurate number of species within the C. podiciferus Species Group, we iteratively compared the results from the mtDNA and nuDNA datasets and prioritized the following: (1) We recognized all 11 currently named species within the C. podiciferus S. G., except that they fail to form separate cluster in the trees (e.g., that samples of species A and B were found mixed, forming a single clade); (2) additional unnamed species can be suggested if they are supported by all the combinations of methods within a dataset (mtDNA or nuDNA); (3) additional unnamed species can be suggested to reconcile discordant results among mtDNA and nuDNA phylogenies; only to avoid synonymizing already named species that were supported in the step 1; and (4) additional unnamed species can be suggested if morphological evidence distinctively supports monophyletic clades found in both mtDNA and nuDNA phylogenies.

Morphological data

Although species descriptions are not the goal of this paper, we collected morphological data when it was available in order to further evaluate our species delimitations. We obtained this information for members of the C. podiciferus Species Group.

Ancestral area reconstruction

We used the nuDNA BEAST time-calibrated tree to infer the ancestral areas. The distribution range of the Craugastor podiciferus Species Group was divided into five areas based on the terrestrial ecoregions of the world proposed by Olson et al. (2001): (A) Costa Rican seasonal moist forest; (B) Isthmian-Atlantic moist forest; (C) Isthmian-Pacific moist forest; (D) Talamancan montane forest; and (E) Choco-Darien moist forest. We used the R package BioGeoBEARS (Matzke, 2014) and implemented the DEC model (Ree & Smith, 2008) within a maximum likelihood framework. Furthermore, a founder-event speciation parameter, J, was added to each of these models. Because no species is distributed over more than four defined areas, we set the maximum number of areas to four.

Results

Phylogeny of the Craugastor podiciferus Species Group

Mitochondrial phylogeny

The resulting mtDNA data matrix included 96 sequences with a total alignment length of 1,216 bp, including gaps (559 bp of 16S and 657 bp of COI). PartitionFinder recommended a GTR+I+G substitution model for 16S and K80+I+G, HKY+I, and GTR+G models for COI codon positions 1, 2, and 3, respectively. The pairwise mitochondrial genetic distances are shown in Table 1.

The mtDNA phylogenies inferred with maximum likelihood and Bayesian analysis from MrBayes (Fig. 2) were almost identical in topology. The clade C. aenigmaticus (red) was consistently inferred as the sister clade to all other lineages within the Species Group. The C. podiciferus clade (blue) contained two well-supported subclades, one including only lineage restricted to the highlands of southwestern Costa Rica and western Panama (C. blairi, C. sagui, and C. zunigai) and another highly structured containing C. podiciferus and six additional lineages from the highlands of Costa Rica. The C. bransfordii group (purple) was paraphyletic with regard to the C. stejnegerianus (green) clade; within the paraphyletic C. bransfordii group, six lineages were well-supported. Craugastor bransfordii was the sister clade to an unnamed lineage from the Caribbean slope of the Talamanca Mountain Range. An unnamed lineage from Panama was consistently supported; C. underwoodi was supported as the sister clade to an unnamed lineage from Caribbean Costa Rica. A sixth unnamed lineage from the Pacific slopes of Costa Rica was supported as the sister clade to all C. stejnegerianus clades. The C. stejnegerianus clade (green) contained six well-supported clades, including C. persimilis, which was consistently supported as the sister clade to all other lineages within the C. stejnegerianus clade, and Craugastor gabbi, which was supported as the sister taxon to the clade formed by C. stejnegerianus + C. rearki. Craugastor stejnegerianus contains two additional unnamed lineages.

Figure 2 Bayesian phylogram derived from MrBayes of the Craugastor podiciferus Species Group based on the 16S and COI mitochondrial DNA gene markers.

Bootstrap proportions are shown above branches. Posterior probabilities (multiplied by 100) from MrBayes and BEAST analyses are shown separated by a slash below the branches (left and right sides, respectively). The scale bar refers to the estimated substitutions per site. The asterisks represent posterior probability values >0.95. Colors represent the following groups identified as clades (Fig. 1): Purple (C. bransfordii ‘clade’), green (C. stejnegerianus clade), blue (C. podiciferus‘clade’), and red (C. aenigmaticus clade). Numbers in parentheses correspond individual ID number provided in Appendix 1. The insert rectangle (left) shows the elevational distribution of lineages within of the C. podiciferus Species Group; black bars correspond with those lineages from Pacific slopes, brown bars correspond to those lineages from Caribbean slopes, and yellow bars correspond to those lineages from both slopes.

The Bayesian MCMC timetree obtained from the mitochondrial dataset using BEAST (Fig. 3) was similar to the mitochondrial phylogenies inferred by RAxML and MrBayes (Fig. 2); the main difference was that the Bayesian analysis weakly supported the monophyly of the C. bransfordii clade (pp = 0.88) versus the paraphyletic C. bransfordii group found in the maximum likelihood and Bayesian analysis with mtDNA dataset. Within the C. bransfordii clade, the unnamed taxon from the Pacific slopes of Costa Rica was supported as the sister taxon to all species within the clade. The next clade to branch off was the clade formed by C. bransfordii + an unnamed lineage from the Caribbean highlands of Costa Rica. Finally, a lineage from Panamá was strongly supported as the sister lineage to the clade formed by C. underwoodi + an unnamed lineage from Caribbean Costa Rica.

Figure 3 Maximum clade credibility tree (left) from the BEAST analysis of Craugastor podiciferus Species Group based on concatenated 16S and COI mitochondrial DNA gene fragments.

Clade colors represent the following: purple (C. bransfordii clade), green (C. stejnegerianus clade); blue (C. podiciferus clade) and red (monotypic C. aenigmaticus clade). Above the branches are shown posterior probabilities (multiplied by 100) from BEAST analysis; the asterisks represent support of >0.95 posterior probability. Numbers in parentheses correspond to individual ID numbers provided in Appendix 1. Comparison of species delimitation results (right) based on the concatenated 16S and COI mitochondrial DNA markers, and for each gene separately (right-most four columns). The six different bar colors correspond to six species delimitation methods used (see ‘Nuclear data’); the bars with same color represents different parameter settings for a given delimitation algorithm. The missing (white) patches in ABGD represent combination of clustering that cannot evaluate in this tree.

In summary, the three phylogenies inferred using mtDNA recovered strong support for three major clades within the C. podiciferus S.G.: a monotypic C. aenigmaticus clade, plus the C. podiciferus and C. stejnegerianus clades. Formal species delimitation results are presented below, while here, we count ‘unnamed lineages’ as the smallest possible number of monophyletic groups that will neither lump nor split named species.

Nuclear phylogeny

The Illumina HiSeq2500 lane generated 66,364,543 reads demultiplexed to 42 samples. The nuclear data matrix contained 7,770 loci and 697,771 characters, with 64.4% missing data (total count of Ns in the matrix). RAxML, MrBayes, and BEAST recovered the same topology (Fig. 4). As in the BEAST analysis of the mtDNA data (see above), the ddRADseq data supported the C. podiciferus Species Group being composed of four major clades, with only minor differences between inference methods with regard to relationships within each of the four clades. Craugastor aenigmaticus (red) was again the sister taxon to all other samples within the Species Group. As with the mtDNA analyses, the C. podiciferus clade (blue) was composed of two subclades, one containing the three highland species, C. blairi, C. sagui, and C. zunigai, and a second highly structured subclade containing C. podiciferus and six additional sample from the highlands of Costa Rica (1,000–2,700 m elevation). The nuclear analysis did not include several populations phylogenetically close to C. podiciferus in the mitochondrial phylogeny.

Figure 4 Maximum clade credibility tree (left) from the BEAST analysis of Craugastor podiciferus Species Group based on 697,771 bp (7,770 loci) from ddRAD dataset.

Clade colors represent the following: purple (C. bransfordii clade), green (C. stejnegerianus clade); blue (C. podiciferus clade) and red (monotypic C. aenigmaticus clade). Bootstrap proportions are shown above branches. Below the branches are shown posterior probabilities (multiplied by 100) from MrBayes analysis (left) and posterior probabilities (multiplied by 100) from BEAST analysis (right). The asterisks represent support of >0.95 posterior probability. Numbers in parentheses correspond to individual ID numbers provided in Appendix 1. Comparison of species delimitation results (right) based on the concatenated nuDNA dataset. The four different bar colors correspond to four species delimitation methods used (see ‘Nuclear data’); the bars with same color represents different parameter settings for a given delimitation algorithm.

The C. bransfordii clade (purple) contains five of the six clades recovered in the mtDNA analyses, C. bransfordii, C. underwoodi, and three additional unnamed lineages, but the relationships within this clade differed from those in the mitochondrial topology. In the ddRADseq tree, Craugastor sp. Panama is sister to the other lineages (essentially from Costa Rica; compare with Figs. 2–3). The C. stejnegerianus clade (green) supported the monophyly of C. gabbi, C. persimilis, and C. rearki. The clade composed of Craugastor stejnegerianus, Craugastor sp. Neilly, and Craugastor sp. Quepos in the mtDNA analysis (Figs. 2–3) was split into three lineages in the nuDNA phylogeny; one was the sister clade to C. rearki, a second one was the C. stejnegerianus clade, and the third one was the sister taxon to gabbi.

Species delimitation

We found considerable differences between the species delimitation analyses performed with the mitochondrial and nuclear datasets and among the various delimitation methods (Figs. 3 and 4). Using some methods, like GMYC and PTP, to look at the mitochondrial data led to the discovery of more than 60 candidate species. However, using others, like mPTP Bayesian delimitation, only 13 candidate species were found. The species delimitation analyses performed on the nuclear data, based on 42 specimens, showed several differences among the species delimitation methods; some (e.g., mPTP Bayesian on MrBayes and RAxML trees) only recognized a single species for the entire complex, whereas others (e.g., mPTP maximum likelihood) identified up to 36 species. We also found incongruences when comparing results for the same clade based on the complete phylogeny versus those based on clade-specific subsets of the data. For example, within the C. stejnegerianus clade, with the GMYCm and PTP methods, we identified six species using the clade-specific phylogeny versus the three identified species using the complete phylogeny. In contrast, within the C. podiciferus clade, we identified 3–5 species using the GMYCm and PTP methods over the complete phylogeny versus 1–2 species identified using a clade-specific phylogeny (Fig. 4). These results highlight the impact of the selection of the initial phylogeny on the results obtained by GMYC, PTP, and mPTP, which are tree-based methods.

According to our priorities for species delimitation, the 11 named species within the C. podiciferus S.G. were supported as monophyletic in the phylogenetic analyses performed on mtDNA and nuDNA datasets. In addition, in the mtDNA analyses, almost all the different methods supported the 11 species as different species; therefore, we validate the 11 named species as distinct.

Following our second criterion, four additional unnamed clades, all within the C. bransfordii clade, were supported by all the different methods in the mtDNA analyses, and therefore, we recognized them as unconfirmed candidate species. These four clades are also supported by our third criterion, which reconciles the mtDNA trees and nuDNA tree. Also, according to our third criterion, two additional clades were recognized as unconfirmed candidate species within the C. stejnegerianus clade to reconcile the mtDNA trees and nuDNA tree. In the mtDNA analyses, Craugastor sp. Neilly and Craugastor sp. Quepos were supported as sister lineages to C. stejnegerianus and delimited as one species (C. stejnegerianus) by several species delimitation methods (Fig. 3). However, in the nuDNA analyses, C. stejnegerianus, Craugastor sp. Neilly, and Craugastor sp. Quepos were not monophyletic (Fig. 4).

Finally, following our fourth criterion, we evaluated the populations contained within the 17 lineages identified above. We found that the nominal species C. podiciferus, as identified by the species delimitation methods, is highly variable, but this variation is fixed in clades; therefore, we suggest that this clade contains at least six additional unconfirmed candidate species. However, these were not delimitated in the mtDNA or nuDNA analyses. We found morphological (Table 2) and acoustic evidence (E Arias, 2024, unpublished data) that supported the distinction of these clades. In addition, some of these clades have been previously suggested as different species (Streicher, Crawford & Edwards, 2009; Arias, Hertz & Parra-Olea, 2019). In summary, by integrating multiple species delimitation methods and our taxonomic expertise in this Species Group, we suggest that it is formed by at least 23 lineages of named species and unconfirmed candidate species.

Table 2 Morphological characters.

Key diagnostic features and variations in secondary sexual characteristics among the lineages of the Craugastor podiciferus clade.

	Head	Venter skin	Accessory palmar tubercles	Heel	Nuptial pads	Vocal slits	
C. podiciferus	Rounded	Smooth	Absent	A projecting tubercle	Absent	Present	
C . sp. Monteverde	Rounded	Smooth	1-2 flatted	Two, a prominent tubercle in the corner of heel	Absent	Present	
C. sp. San Gerardo	Rounded	Areolate	2-4 projecting	1-3 low not-projecting	Present	Present	
C. sp. Pico Blanco	Rounded	Areolate	2-3 flatted	Small granules	Absent	Absent	
C . sp. Siola	Rounded, broad	Smooth with scattered granules	Absent	Fleshy calcar	ND	ND	
C. sp. Chumacera	Subelliptical	Smooth	2-4 not-projecting	1-3 low not-projecting	Absent	Present	
C . sp. Fila Costeña	Rounded	Areolate	2-4 flatted	1-3 low not-projecting	Absent	Absent	

Biogeography

Ancestral area reconstruction assuming the BEAST tree for ddRADseq data inferred a Talamancan origin for the C. podiciferus Species Group during the middle Miocene (Fig. 5). A Talamancan origin for the C. podiciferus clade was also supported, with a dispersal event to Isthmian-Pacific moist forest (lowland) and another dispersal event to Costa Rican seasonal moist forest. Our data support the origin of the C. stejnegerianus + C. bransfordii clade in Isthmian-Atlantic moist forest, with five independent dispersal events from the Isthmian-Atlantic moist forest to Talamancan montane forest during the Pliocene-Pleistocene and three dispersal events to Isthmian-Pacific moist forest, explaining the current patterns of distribution. The two basal clades, Craugastor aenigmaticus clade (in red) and C. podiciferus clade (in blue), are restricted to highlands (1,000–2,700 m), yet all 23 lineages have populations above 750 m in the Talamancan montane forest (Figs. 1 and 2). Lineages found in the highlands have narrower elevational ranges, except C. podiciferus and C. blairi (Figs. 1 and 2); all lineages distributed below 1,000 m have altitudinal ranges greater than 750 m in extent, but six (Craugastor sp. Chumacera, Craugastor sp. Fila Costeña, Craugastor sp. Pico Blanco, Craugastor sp. San Gerardo, Craugastor sp. Siola, and Craugastor sp. Vereh) out of 15 lineages restricted to highlands have altitudinal ranges smaller than 250 m. In addition, the highlands of the Pacific versant are more structured; in the Pacific versant, nine lineages (C. blairi, C. gabbi, C. sagui, C. zunigai, Craugastor sp. Chumacera, Craugastor sp. Fila Costeña, Craugastor sp. Pico Blanco, Craugastor sp. Quebradas, and Craugastor sp. San Gerardo) are distributed exclusively over 1,000 m. However, in the Atlantic versant, only five lineages (C. podiciferus, C. underwoodi, Craugastor sp. Monte Verde, Craugastor sp. Siola, and Craugastor sp. Vereh) are restricted to highlands.

Figure 5 Ancestral area reconstruction from BioGeoBEARS using the DEC+J model, and the tree derived from the Bayesian analyses of ddRAD dataset.

Most likely biogeographic areas are shown in the pies, and the colors in the squares indicate the current species distribution. Numbers in parentheses by sample names correspond to individual ID numbers provided in Appendix I. The map shows the location of the areas used. The DEM used in this study was obtained online from CGIAR-CSI SRTM website: http://srtm.csi.cgiar.org.

Discussion

Systematics and biogeography of the C. podiciferus Species Group

The highlands of the ICA played an important role in the diversification of several groups of vertebrates (García-París et al., 2000; Savage, 2002; Castoe et al., 2009; Boza-Oviedo et al., 2012; Duellman, Marion & Hedges, 2016; Arias, Chaves & Parra-Olea, 2018; Arias, Hertz & Parra-Olea, 2019), and given that the basal clades within the nuDNA and mtDNA trees are restricted to the highlands of the ICA (Figs. 2 and 5), the C. podiciferus Species Group appears to follow this pattern. We used the results of Streicher, Crawford & Edwards (2009) as our prior for the time of origin for the most recent common ancestor (MRCA) of the C. podiciferus Species Group between 16.8 and 27.8 Mya when the ancestor dispersed from Nuclear Central America to the ICA as the latter emerged as a peninsula at its current position (Montes et al., 2015). Our posterior estimates matched our priors, indicating that our data are at least consistent with the four major clades within the C. podiciferus Species Group having diverged during the Miocene. Diversification within the ICA during the Miocene and Pliocene has been found in other northern lineages of anurans (Duellman, Marion & Hedges, 2016), pitvipers (Castoe et al., 2009), and freshwater fishes (Říčan et al., 2013).

All identified lineages of the C. podiciferus Species Group are distributed from eastern Honduras to eastern Panama in an area smaller than 40,000 km2. Craugastor aenigmaticus is restricted to Talamanca montane forest, occurring from 2,330–2,700 m, the highest elevational distribution for any species in this group (Arias, Chaves & Parra-Olea, 2018). Craugastor aenigmaticus the sister taxon to the rest of the Species Group, and according to our ancestral area reconstruction, the ancestor was distributed in the Talamanca montane forest and diversified to lower elevations (Fig. 5). The C. podiciferus clade, as suggested by Streicher, Crawford & Edwards (2009), possibly diversified due to climatic fluctuations that isolated the suitable habitat on peaks in the mountain ranges. The C. bransfordii + C. stejnegerianus clade contains species ranging from sea level to 1,600 m elevation. The C. bransfordii clade diversified mainly on the Caribbean slopes of Nicaragua, Costa Rica, and Panama. Only one lineage was found on the Pacific slopes of Costa Rica. No obvious barriers separate the species of the C. bransfordii clade, except for paleoclimatic differences. The C. stejnegerianus clade contains more species on the Pacific slope of Costa Rica, with only two species distributed on the Caribbean slopes of Nicaragua, Costa Rica, and Panama. Possibly, the drier conditions found on the Pacific slope of Costa Rica during the Pleistocene (Crawford, Bermingham & Polanía, 2007) allowed for the diversification of this clade in this region, isolating the ancestors in more mesic areas.

In the lowlands, the climatic oscillations and the fluctuations in sea level during the Pliocene may have fragmented distribution ranges, isolating populations and restricting gene flow. These climatic fluctuations mainly affected the Pacific slopes (Savage, 1966; Crawford, Bermingham & Polanía, 2007). If these fluctuations were long enough, they could explain the high genetic structure found here, with species having narrow elevational and latitudinal ranges (Figs. 2 and 5). The Pacific slopes have more highland lineages than the Caribbean slope. This could be due to the climatic heterogeneity found there; for example, the mean annual precipitation varies from ∼1,800 mm in the Northern Pacific to nearly 5,000 mm in the Southern Pacific in less than 400 lineal km (Savage, 1966; Coen, 1991). Recently, this heterogeneity in the Pacific slopes has been suggested to promote speciation in three species of Anolis lizards distributed along the Central and South Pacific (Chaves et al., 2023). In contrast, the Caribbean slopes have been presumed to be more homogeneous in temperature and precipitation (A. García-Rodríguez, E. Arias, J.A. Velasco, G. Parra-Olea, 2025, unpublished data) although fluctuations in sea level (Bagley & Johnson, 2014) may have promoted diversification of lineages in that slope.

Species delimitation and taxonomic comments

Seven named species and candidate species (C. aenigmaticus, C. blairi, C. persimilis, C. podiciferus, C. sagui, C. zunigai, and Craugastor sp. Panama) were supported as distinct evolutionary lineages in most analyses. Nevertheless, we found substantial differences in delimitation results between mitochondrial and nuclear analyses and among methods. We identified 23 lineages (Figs. 6 and 7), including named species and unconfirmed candidate species (Appendix 1). Our approach aims to minimize taxonomic instability while recognizing the full biodiversity contained within this group. Below are details of the major clades and their species.

Figure 6 Geographic distribution of Craugastor podiciferus Species Group.

Geographic distribution of all named species and candidate species within of Craugastor podiciferus Species Group in Costa Rica and western Panama. The mountains of southeastern Costa Rica are referred to as the Talamanca Mountain Range in the text. The purple shapes correspond to the C. bransfordii clade; green shapes = C. stejnegerianus clade; blue shapes = C. podiciferus clade; red shapes = C. aenigmaticus clade. The DEM used in this study was obtained online from CGIAR-CSI SRTM website: http://srtm.csi.cgiar.org.

Figure 7 Photographs in life of Craugastor podiciferus species complex.

Photographs in life of (A) Craugastor aenigmaticus (UCR 22961) from Cerro Arbolado, Puntarenas, CR, (B) C. blairi (SMF 104032) from Fortuna, PA, (C) C. sagui (SMF 104018) from La Nevera, PA, (D) C. zunigai (UCR 20389) from Potrero Grande, Puntarenas, CR, (E-F) C. podiciferus (UCR 23155, 23159) from Caribbean slopes of Cerro Kamuk, Limón, CR, (G) Craugastor sp. Monte Verde (UCR 24613) from Monte Verde, Puntarenas, CR, (H) Craugastor sp. San Gerardo (CRARC 0247) from San Gerardo, Guanacaste, CR, (I) Craugastor sp. Fila Costeña (UCR 23028) from Quebradas, San José, CR, (J) Craugastor sp. Pico Blanco (UCR 24466) from Escazú, San José, CR, (K) Craugastor sp. Chumacera (UCR 23011) from Chumacera, San José, CR, (L) Craugastor sp. Siola (UCR 23169) from Siola, Limón, CR, (M) C. bransfordii from Siquirres, Limón, CR, (N) Craugastor sp. Fila Carbon (UCR 23127) from Amubri, Limón, CR, (O) C. underwoodi from Cascajal, San José, CR, (P) Craugastor sp. Quebradas from Fila Costeña, Puntarenas, CR, (Q) Craugastor sp. Vereh (UCR 23040) from Vereh, Cartago, CR, (R) Craugastor sp. Panama (SMF 104010) from Rambala. PA, (S) C. stejnegerianus (UCR 22976) from Palmar Norte, Puntarenas, CR, (T) C. gabbi (UCR 22998) from San Vito, Puntarenas, CR, (U) C. persimilis from Siquirres, Limón, CR, (V) C. rearki from Siquirres, Limón, CR, (W) Craugastor sp. Neilly (UCR 22985) from Río Claro, Puntarenas, CR, and (X) Craugastor sp. Quepos (UCR 24612) from Montes de Oca, San José CR. Photos by E. Arias (A, E, F, G, I, J, K, L, N, Q, S, T, W, and X), Andreas Hertz (B,C, and R), Eduardo Boza-Oviedo (D and O), Brian Kubicki (H, M, U, and V), and Raby Nuñez (P).

Craugastor aenigmaticus

This species (Fig. 7A) was consistently supported as a distinct evolutionary lineage in all mtDNA and nuDNA analyses. This species is notably separated from other C. podiciferus Species Group members by mean uncorrected genetic distances greater than 13.3% in 16S and 19.3% in COI (Table 2).

The Craugastor podiciferus clade

The species C. blairi (Fig. 7B), C. sagui (Fig. 7C), and C. zunigai (Fig. 7D) are each confirmed as distinct species from the topotypic C. podiciferus by several delimitation methods. The mitochondrial and nuclear phylogenies placed them as a monophyletic group sister to C. podiciferus + candidate species (Figs. 2–4). They are separated from each other by mean uncorrected genetic distances of at least 3.4% in 16S and 17.3% in COI (Table 2). Craugastor blairi, C. sagui, and C. zunigai are allopatric and distributed latitudinally in southwestern Costa Rica and western Panama (Arias, Hertz & Parra-Olea, 2019). Craugastor blairi corresponds to Craugastor sp. B of Crawford & Smith (2005) and clade G of Streicher, Crawford & Edwards (2009).

The number of species masked under the name C. podiciferus remains unclear. We consistently identified several candidate species with mitochondrial sequence data, but only some were included in the species delimitation analyses based on nuDNA data. Streicher, Crawford & Edwards (2009) performed a phylogenetic analysis for this clade, suggesting that it was composed of multiple taxa. Of the seven candidate species in our mitochondrial phylogeny, only three were included in the nuclear dataset, which were supported as separate species in BPP analyses.

We included samples from the type locality of C. podiciferus (Figs. 7E–7F). The type locality was discussed by Savage (1970) and Arias & Chaves (2014), who later corrected the type locality to the Caribbean slope of Cerro Kamuk. Here, we restrict C. podiciferus to the populations of Cordillera Volcánica Central from Costa Rica and the Cordillera de Talamanca of Costa Rica and western Panama. In the Cordillera de Talamanca, C. podiciferus is restricted to the Caribbean slopes.

Based on mitochondrial data, we identified seven lineages within the name C. podiciferus, some of which also differ morphologically (Table 2 and E Arias, 2024, unpublished data). These lineages are separated from each other by mean uncorrected genetic distances of 2.0–10.5% in 16S and 7.3–23.4% in COI (Table 2). Thus, we suggest that these seven lineages represent separate species (C. podiciferus sensu stricto and 6 unconfirmed candidate species). Streicher, Crawford & Edwards (2009) included samples from four of these taxa: C. podiciferus sensu stricto corresponds to clades C and D of Streicher, Crawford & Edwards (2009), Craugastor sp. Monte Verde (Fig. 7G) corresponds to clade A of Streicher, Crawford & Edwards (2009), a species restricted to the Cordillera de Tilarán and Cordillera Volcánica Central. Craugastor sp. San Gerardo (Fig. 7H) corresponds to clade B of Streicher, Crawford & Edwards (2009), a species distributed on the Cordillera de Tilarán and Volcánica Central. In Monte Verde Craugastor sp. Monte Verde and Craugastor sp. San Gerardo are sympatric (Streicher, Crawford & Edwards, 2009). Sympatry between C. podiciferus and Craugastor sp. Monte Verde near Zarcero was also recorded by Streicher, Crawford & Edwards (2009). Craugastor sp. Fila Costeña (Fig. 7I) corresponds to clades E and F of Streicher, Crawford & Edwards (2009), a species restricted to southern Pacific Costa Rica. The remaining unconfirmed candidate species Craugastor sp. Siola (Fig. 7L), Craugastor sp. Pico Blanco (Fig. 7J), and Craugastor sp. Chumacera (Fig. 7K) have not been included in any previous work. Craugastor sp. Pico Blanco is known only from one site in Valle Central. Craugastor sp. Chumacera is known only from one site on the Pacific slope of the Cordillera de Talamanca, and Craugastor sp. Siola is known from a single population on the Caribbean slope of the Cordillera de Talamanca.

Streicher, Crawford & Edwards (2009) estimated a time to MRCA of the C. podiciferus clade of between 4.70 and 8.18 Ma. We hypothesize that the lack of support for the distinctness of these taxa in some species delimitation methods (e.g., mPTP, BPP) may reflect the fact that they are recently derived, as shown by their lower genetic divergence. However, morphological and acoustic evidence (E Arias, 2024, unpublished data) suggests that they are nonetheless on evolutionarily independent trajectories and therefore should be recognized as separate species.

The Craugastor bransfordii clade

The nuclear phylogeny supports the monophyly of the C. bransfordii group but the RAxML and MrBayes of the mitochondrial analyses do not. We suggest that the C. bransfordii clade is composed of six separate species, only five of which were included in the nuclear analysis. These six lineages are separated from each other by mean uncorrected genetic distances between 4.1–11.3% in 16S and 9.5–17.3% in COI (Table 2). Craugastor bransfordii (Fig. 7M) samples included a specimen (UCR 20559, ID 78) collected near the type locality, San Juan River, on the border between Costa Rica and Nicaragua. Craugastor bransfordii is distributed from northern Nicaragua to central Caribbean Costa Rica. The nominal species C. polyptychus was described with specimens from the same type locality as C. bransfordii and in the same publication (Cope, 1886). It was later recognized as a synonym of C. bransfordii (Savage & Emerson, 1970; Miyamoto, 1983) until Savage (2002) resurrected this name and assigned it to specimens from Caribbean Costa Rica but noted that further taxonomic work is needed to clarify the status of the species. Since only one lineage within the C. bransfordii clade was found in northern Costa Rica and southern Nicaragua, we suggest that C. polyptychus should be referred to as a junior synonym of C. bransfordii. Craugastor sp. Fila Carbón (Fig. 7N) corresponds –in part–to C. polyptychus of Savage (2002), but since the type locality of C. polyptychus is outside of the range of our Craugastor sp. Fila Carbón, this unconfirmed candidate species, will require a new formal description and name. Based on all but one (mPTPbs) of the 24 mtDNA species delimitation analyses, including genetic distance, this lineage represents an unconfirmed candidate species distributed in southeastern Caribbean Costa Rica and western-most Panama.

Craugastor underwoodi (Fig. 7O) includes specimens from Vázquez de Coronado (ID 82), near the type locality. This species is distributed in the premontane forest of the Cordillera de Guanacaste, Cordillera de Tilarán, Cordillera Volcánica Central, and the northern edge of the Cordillera de Talamanca. Craugastor sp. Quebradas (Fig. 7P) includes specimens from the only locality known on the Pacific slope for a member of the C. bransfordii clade. We consider that Craugastor sp. Quebradas represents a separate species due to its allopatric distribution and the large genetic distances from all other samples, with 15.3% or higher in COI (Table 1). Craugastor sp. Vereh (Fig. 7Q) includes specimens from two localities in the premontane forest in the central Caribbean of Costa Rica. Finally, Craugastor sp. Panama (Fig. 7R) is composed of specimens from Panama that were assigned to C. bransfordii (Leenders, 2016), but based on the mitochondrial and nuclear phylogenies, this clade from Panama is not closely related to C. bransfordii. We suggest that these populations from Panama represent an unconfirmed candidate species.

The Craugastor stejnegerianus clade

Within the C. stejnegerianus clade, large differences were found between the mitochondrial and nuclear phylogenies, mainly in the relationships between the recently described C. gabbi and C. stejnegerianus. We suggest that the C. stejnegerianus clade is composed of six independent lineages. Craugastor persimilis (Fig. 7U) is represented here by specimens from the type locality, Suretka, Cantón de Talamanca (ID 43). Craugastor persimilis is distributed on the central and southern Caribbean slopes of Costa Rica (Fig. 6).

The specimen from Honduras (ID 58; USNM 559393) was previously examined and identified as C. lauraster by (McCranie, 2006) (coauthor of the original publication describing C. lauraster). This specimen from Honduras is grouped with specimens from Nicaragua (ID 51–52), that were tentatively referred to as C. lauraster, and those specimens from the central Caribbean coast of Costa Rica (ID 50, 55–57). These specimens from Costa Rica agree morphologically with C. rearki (Taylor, 1952), synonymized earlier with C. bransfordii by Savage & Emerson (1970). Craugastor rearki is diagnosable by its (1) chin, chest, and venter nearly immaculate white; (2) dorsum with 3-4 pairs of longitudinal folds; (3) head nearly round to slightly subelliptical; (4) Finger I equal or slightly shorter that Finger II; and (5) absence of nuptial pads. The above-mentioned characteristics match the specimens from Costa Rica and are included in the description of C. lauraster. We included one specimen from Siquírres (ID 50), near the type locality of C. rearki, and one specimen from Pococí (ID 55), a locality of paratypes of C. rearki. We suggest that the name C. rearki (Fig. 7V) should be resurrected to include populations from the Caribbean of Costa Rica, Nicaragua, and Honduras, and the newer name, C. lauraster, should be referred to as a junior synonym of C. rearki.

There are considerable differences between the mitochondrial and nuclear phylogenies on the relationship of C. stejnegerianus from the Pacific slopes and the other members of the clade. Arias et al. (2016) supported the distinctiveness of C. stejnegerianus (Fig. 7S) and C. gabbi (Fig. 7T) based on mitochondrial phylogeny, morphology, and ecological preferences. Cossel Jr et al. (2019) recorded differences in advertisement calls among C. gabbi, C. stejnegerianus, and Craugastor sp. Quepos (referred to as C. stejnegerianus Northern). The mitochondrial phylogeny clusters C. stejnegerianus with Craugastor sp. Neilly and Craugastor sp. Quepos, whereas in the nuclear phylogeny, C. stejnegerianus sensu stricto is the sister taxon to C. gabbi + Craugastor sp. Neilly. Based on both molecular analyses, we recognize Craugastor sp. Neilly (Fig. 7W) to conciliate its different phylogenetic positions among mtDNA analysis (related to C. stejnegerianus) and nuDNA analysis (sister taxon to C. gabbi), this taxon is restricted to the southeast Pacific of Costa Rica. Similarly, we recognize Craugastor sp. Quepos (Fig. 7X), to conciliate its different phylogenetic position among mtDNA analysis (sister taxon to C. stejnegerianus) and nuDNA analyses (sister taxon to C. rearki); this taxon is distributed in the central Pacific and Central Valley of Costa Rica. These three species are separated from C. persimilis, C. gabbi, and C. rearki by mean uncorrected genetic distances between 4.2–9.0% in 16S and 11.6–17.2% in COI (Table 1).

Conclusions

The diversity within the Craugastor podiciferus Species Group is vastly underestimated, as revealed by the presence of several undescribed species recovered from the phylogenetic and species delimitation analyses. An exhaustive morphological review of the genetic lineages may show morphological characteristics that would allow for the differentiation of the molecular lineages. Comprehensive studies are needed on habitat use, acoustics, behavior, and other data to corroborate and better understand the taxonomy of all lineages revealed here.

Based on our mitochondrial and nuclear analyses, morphological evidence, and previous information we recovered 23 lineages, 11 with names and 12 unconfirmed candidate species. Based on our results, we propose the following changes:

• We restrict C. podiciferus to populations of Cordillera Volcánica Central from Costa Rica and the Cordillera de Talamanca of Costa Rica and western Panama. In the Cordillera de Talamanca, C. podiciferus is restricted to the Caribbean slopes.

• Craugastor polyptychus is referred to as a junior synonym of C. bransfordii.

• Craugastor rearki is resurrected to include wide-ranging populations from the Caribbean versant of Costa Rica, Nicaragua, and Honduras.

• Craugastor lauraster is referred to as a junior synonym of the older name, C. rearki.

• Finally, we want to highlight the need to continue exploring remote areas in the ICA, especially in the Talamanca Mountain Range. The fieldwork performed in this area has resulted in the discovery of several new species or new records for the region. Therefore, more fieldwork and laboratory work are necessary to improve the knowledge of biodiversity in this region to perform informed strategies of conservation.

Supplemental Information

Supplemental Information 1 Voucher information, locality data and Genbank sequences for 110 species

Institutional voucher number (or field collector number) and locality information for the specimens used in molecular phylogenetic analyses. Museum collection acronyms follow Frost (2023), with the addition of the Círculo Herpetológico de Panamá (CH). Field collector numbers refer to Andrew J. Crawford (AJC), Erick Arias (EAP), and the Costa Rica Amphibian Research Center private collection (CRARC).

Supplemental Information 2 Results of multiples replicates of the seventh step of the software pipeline ipyrad v0.5.15

Results of multiples replicates of the seventh step of the software pipeline ipyrad v0.5.15 (Eaton, 2014), to determine the optimal value for the parameters: clustering parameter, maximum numbers of SNPs allowed in a locus, maximum number of insertions/deletions allowed in among-sample clusters, and the maximum proportion of samples allowed to share a heterozygous site. (A) Variation in the proportion of clusters with 1, 2, and ¿2 alleles retrieved with different similarity thresholds; (B) variation in the number of retrieved loci with different maximum numbers of SNPs in a final locus; (C) variation in the number of retrieved loci with different maximum numbers of insertions/deletions in across-sample clusters (c); (D) variation in the number of retrieved loci with different maximum proportions of samples with a shared heterozygous site. The different line colors in a represent different numbers of alleles/cluster, as indicated below the graph.

Supplemental Information 3 Mt and Nuc sequences generated for this study

We thank Laura Márquez-Valdelamar and Andrea Jiménez-Marín for their laboratory assistance; Federico Bolaños for the use of specimens from the Museo de Zoología of the Universidad de Costa Rica; Omar Zúñiga, Olmer Cordero, Justo Layam Gabb, and Xavier Baltodano provided valuable assistance in the field during the expeditions. Brian Kubicki for its valuable assistance and for donating tissue samples. Adrián García-Rodríguez provided valuable mitochondrial sequences and comments on earlier versions of the manuscript, which was also improved due the kind suggestions of Gerardo Chaves. This paper constitutes a partial fulfillment of the Graduate Program in Biological Sciences (Posgrado en Ciencias Biológicas) for EA.

Additional Information and Declarations

Competing Interests

Author Contributions

Field Study Permissions

DNA Deposition

Data Availability

Gabriela Parra Olea is an Academic Editor for PeerJ.

Erick Arias conceived and designed the experiments, performed the experiments, analyzed the data, prepared figures and/or tables, authored or reviewed drafts of the article, and approved the final draft.

Andrew J. Crawford conceived and designed the experiments, authored or reviewed drafts of the article, and approved the final draft.

Andreas Hertz conceived and designed the experiments, authored or reviewed drafts of the article, and approved the final draft.

Gabriela Parra Olea conceived and designed the experiments, analyzed the data, authored or reviewed drafts of the article, and approved the final draft.

The following information was supplied relating to field study approvals (i.e., approving body and any reference numbers):

The Costa Rican Ministry of Environment and Energy (MINAE) provided the corresponding scientific collection permits for this research (SINAC-SE-GAS-PI-R 007-2013 and 59-2015). Collecting permits for Panama SE/A-30-08, SC/A-8-09, SC/A-28-09, and SC/A-21-10, as well as the corresponding exportation permits, were issued by the Ministerio de Ambiente (MiAmbiente), Panama City, Panama. Collecting permits for Nicaragua No. 006–062009 was issued by Ministerio del Ambiente y los Recursos Naturales, Managua, Nicaragua.

The following information was supplied regarding the deposition of DNA sequences:

The GenBank sequences for 110 species are available in the Supplemental File.

The following information was supplied regarding data availability:

The data are available in the Supplemental File.

The 16S sequences are available at NCBI: OR419510, MK211615, MK211616, MK211617, MK211623, MK211624, MK211625, MK211626, MK211628, MK211627, MK211629, MK211633, MK211640, MK211641, MK211639, MK211637, MK211636, MK211638, MK211632, MK211635, MK211634, MK211642, MK211631, EF562367, MK211647, MK211646, EF562372, EF562343, EF562371, MK211645, EF562374, EF562349, MK211644, MK211643, MK211630, OR419511, KT950271, KT950272, MK211609, OR419512, KT950293, OR419513, OR419514, MK211608, OR419515, OR419516, OR419517, OR419518, OR419519, KU323364, MK211607, KT950283, KT950284, KT950292, KT950281, KT950280, OR419520, OR419521, OR419522, OR419523, OR419524, OR419525, OR419526, OR419527, MK211610, OR419528, OR419529, KT950295, MK211611, MK211612, OR419530, OR419531, MK211613, OR419532, MK211614, OR419533, OR419534, OR419535, KR863145, KR863147, KR863146, OR419536, FJ784358, FJ784427, FJ784339, FJ784376, FJ784481, FJ784496, OR419537, OR419538, OR419539, OR419540.

The COI sequences are available at NCBI: OR420802, MK211577, MK211578, MK211579, MK211580, MK211581, MK211582, MK211584, MK211583, MK211585, MK211589, MK211596, MK211597, MK211595, MK211593, MK211592, MK211594, MK211588, MK211591, MK211590, MK211587, MK211605, MK211604, MK211598, MK211603, MK211606, MK211599, MK211602, MK211601, MK211600, MK211586, OR420803, MK211567, MK211568, MK211569, OR420804, MK211570, OR420805, OR420806, MK211565, OR420807, OR420808, OR420809, OR420810, OR420811, MK211566, MK211563, OR420812, MK211564, OR420813, OR420814, OR420815, OR420816, OR420817, OR420818, OR420819, OR420820, OR420821, OR420822, OR420823, MK211572, OR420824, OR420825, MK211571, MK211573, MK211574, OR420826, MK211575, OR420827, MK211576, OR420828, OR420829, OR420830, OR420831, KR862890, KR862892, KR862891, OR420832, FJ766630, FJ766628, FJ766631, FJ766629, FJ766627, FJ766626, OR420833, OR420834, OR420835, OR420836, OR420837.

The ddRAD are available at NCBI: SAMN43761706, SAMN43761707, SAMN43761708, SAMN43761709, SAMN43761710, SAMN43761711, SAMN43761712, SAMN43761713, SAMN43761714, SAMN43761715, SAMN43761716, SAMN43761717, SAMN43761718, SAMN43761719, SAMN43761720, SAMN43761721, SAMN43761722, SAMN43761723, SAMN43761724, SAMN43761725, SAMN43761726, SAMN43761727, SAMN43761728, SAMN43761729, SAMN43761730, SAMN43761731, SAMN43761732, SAMN43761733, SAMN43761734, SAMN43761735, SAMN43761736, SAMN43761737, SAMN43761738, SAMN43761739, SAMN43761740, SAMN43761741, SAMN43761742, SAMN43761743, SAMN43761744, SAMN43761745, SAMN43761746, SAMN43761747.

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
