# Peer review of "Deep cryptic diversity in the Craugastor podiciferus Species Group (Anura: Craugastoridae) of Isthmian Central America revealed by mitochondrial and nuclear data"

_PeerJ, doi:10.7717/peerj.18212_

## Round 0.1 · original submission · Major Revisions

Please, deal with the relevant issues stated especially by reviewers 2 and 3, and provide a fully revised version of the manuscript.

·

Basic reporting

The manuscript comply the structure of an article to be published in the journal. The ms has a professional structure.

Experimental design

Original primary research within Aims and Scope of the journal.

Validity of the findings

The authors, shown new data and give insight of the taxonomic status of the species, promoting the needs of new works to be done.

·

Basic reporting

no comment

Experimental design

no comment

Validity of the findings

no comment

Additional comments

Review of Deep cryptic diversity in the Craugastor podiciferus species group (Anura: Craugastoridae) of Isthmian Central America revealed by mitochondrial and nuclear data by Arias et al.

Summary: I sincerely apologize to the authors for the delay in returning my review. I read this manuscript with great interest because I have been following these authors' work and I was also an author on a previous paper studying one of the focal clades (C. podiciferus). The manuscript describes mitochondrial (16S, CO1) and nuclear (ddRADseq) data that are applied with the goals of delimiting species-level diversity as well as estimating the ancestral geographic origin of the C. podiciferus species group (sensu Hedges et al. 2008). I thought the paper was overall well-written (though it requires clarification in a few places). I also found the paper to be quite thought-provoking and congratulate the authors on a very interesting study. Below I have provided some major suggestions for improving the study and also some minor comments to clarify/improve the text.

Major suggestions:

1. Non-molecular evidence and species boundaries: A major goal of this paper is to delimit species within the C. podiciferus species group. The authors make multiple references to unpublished morphological and acoustic data that may (or may not) support the results of the mtDNA and nDNA results presented in the paper. In particular, data from acoustics and morphology seem particularly relevant given (1) the reported mito-nuclear discordance and (2) some of the low support values for clades of 'undescribed' species in the mtDNA dataset (i.e. 70 for Monte Verde species, and 69 for San Gerardo species; Fig 3). I found myself wondering why the morphological and acoustic data were not included because on lines 357-358 it is implied that morphological data are going to be used to back up molecular delimitation results. If it is possible to include these data in a revision, I think the suggested delimitation scheme would be strengthened.

2. Putting names on tips: How were voucher specimens assigned to species? As the authors point out in the abstract - many of the focal species have intraspecific polymorphism - so how exactly do we know which clades correspond to which species? Geographic distribution? Non-color based morphological characters? I trust the taxonomic expertise of the authors, it just makes it very difficult to follow along as a reader when it isn't clear if the authors a priori assigned taxonomy or did that as a post hoc procedure. I also suspect that the authors have seen most (if not all) of the relevant type material for this group (e.g. Arias et al. 2019; Amphibian and Reptile Conservation), so some context for how taxonomic assignment occurred would be appreciated.

Minor suggestions:

Introduction

Line 80: Streicher et al. (2009) referred to this as clade G and Craugastor sp. A (not Craugastor sp. B). Also, I think this 'undescribed species' turned out to be C. blairi according to Arias et al. (2019) - so this should be mentioned here to make things clear.

Methods

Line 139: parentheses missing from (Gene Codes Corp.

Lines 169-170: An ultrametric tree is a rooted tree with edge lengths where all leaves are equidistant from the root. Unless using a 'strict molecular clock' approach where all branches are evolving at the same rate, the tree won't be ultrametric. Related comment - what about calibration points?

Lines 191-192: If there are only 48 samples why do you need a 96 well plate? Please clarify.

Lines 197-198: This sentence seems to imply the samples were cleaned with a Qubit 2.0 fluorometer, which can only quantify amounts of DNA/RNA (not clean them).

Lines 247-249: How was this missing data threshold established?

Lines 266-272: Were these the same calibration procedures used in the mtDNA divergence time analysis? If so, it seems like this description should appear earlier in the methods for clarity.

Line 281: Why were only the partial phylogenies used with BPP?

Line 313: Why was ABGD only performed on the mtDNA dataset? Seems inconsistent with the other species delimitation methods.

Line 325: Again, please check that using the term ultrametric is correct here.

Line 348: How exactly is it known that the different species delimitation methods will not agree? Has there been previous work to demonstrate this? If so, that should be cited here.

Lines 348-358: This paragraph confused me. I get the general idea that is being presented, but I think it could be worded more clearly. Also, what morphological data are being used in point 4 and how exactly are these being used to support the monophyly of clades?

Line 351: Please define S.G. for readers.

Line 404: '...recovered strong support for a monophyletic C. podiciferus species group relative to our outgroup, C. loki...' A single outgroup can't be used to test monophyly of the ingroup because you can reroot any tree so that a single taxon makes the other taxa 'monophyletic'. See what I mean? On a related note, why weren't C. rhodopis and C. occidentalis also included as outgroups? There is mtDNA of (at least) 16S for both of these species online. I don't think this is essential to do for the revision, just curious.

Line 453: 'We validated the 11 species as different' - but you previously wrote 12 species in this paragraph. Is it 11 or 12?

Lines 466-468: 'We found morphological and acoustic evidence (E Arias, 2023, unpublished data) that supported the distinction of these monophyletic clades.' This is an very interesting statement, but why aren't these morphological and acoustic data included in the present paper? They seem highly relevant to species delimitation.

Lines 560-562: I think this statement: 'Craugastor blairi corresponds to Craugastor sp. B of Crawford & Smith (2005) and clade G of Streicher, Crawford & Edwards (2009).' would be better in the Introduction along with a citation for Arias, Hertz & Parra-Olea (2019) who reported the identity of C. blairi.

Lines 565-567: Streicher et al. (2009) reported six clades, not six species. Because of the syntopic occurrence of haplogroups I think incomplete lineage sorting of mtDNA haplotypes can't be ruled out as an explanation for the pattern - so we were careful not to call them 'species'. It would be fair to write that you recovered similar results to Streicher et al. (2009) and that you consider this to be evidence of multiple species. On a related note, it doesn't seem like you have enough nuclear data to assess whether the syntopic occurrence of divergent mtDNA types is explained by different species or diverse mtDNA in a single interbreeding population. Please correct me if I am wrong.

Lines 576-577: 'Based on mitochondrial data, we identified seven lineages within the name C. podiciferus, some of which also differ morphologically (E Arias, 2023, unpublished data).' Again, I wonder why these morphological data can't be included in this manuscript to support (or challenge) molecular clades being equivalent to species.

Lines 586-588: What about the results of Streicher et al. (2009) that their clades A and B were both observed in northern Puntarenas (I think Monteverde)? See 'Zone of Sympatry' in their Figure 1.

Lines 600-602: 'However, morphological and acoustic evidence (E Arias, unpublished data) suggests that they are nonetheless on evolutionarily independent trajectories and therefore should be recognized as separate species.' For the final time (I promise), given that one of the major goals of the manuscript is 'species delimitation' it is unclear why these unpublished data would not be presented and used to better delimit diversity amongst these frogs.

Lines 630: I wonder about the allopatry statements here related to C. underwoodi and the putative undescribed species - in other words looking at the sampling map Fig. 6, there appear to be only three localities where C. underwoodi was sampled, so is it safe to conclude the populations/species are allopatric? Or could they be connected genetically by unsampled populations?

Figures: In general, the figures look great to me!

Figure 3: I am confused by this part of the figure caption: 'The missing (white) patches in ABGD represent combination of clustering that cannot evaluate in this tree.' Is it combinations of clustering or that the taxa are missing 16S or CO1 data? Sorry I have not used the ABGD method, so a little more context for what is meant here would be helpful (for me and for other readers that have not used ABGD).

I greatly enjoyed reading this manuscript and hope that my comments will be helpful for revising the paper. I am happy for the authors to contact me directly for clarification on my comments and suggestions.

Again, I congratulate the authors on an exciting paper!

Jeff Streicher
[email protected]

Reviewer 3 ·

Basic reporting

In general, the English language used es mostly understandable and unambiguous. However, several sentences throughout the manuscript are not entirely well written and are hard to follow, and there is also a number of small details that need to be corrected. Thus, the English needs improvement. Not being a native English speaker, I leave this to the better judgment of other colleagues.

The Introduction provides context and the literature is well referenced and relevant. However, some relevant literature concerning the methods was not included.

In general, the structure seems to conform to PeerJ standards; however, the Material & Methods section should be considerably more concise, since some methods used are now rather standard and do not need to be described in detail; also, some abbreviations need to be eliminated (comments on manuscript file).

The figures are relevant, of good quality, and in general well labelled; however, as in the body of the document, the English language in the description of some figures needs to be improved.

The data was deposited in GenBank; accession numbers are provided for mitochondrial data, though they are pending for RADseq data. The deposition information is noted in the Material & Methods section and provided in an Appendix. I can access the deposited data with an accession number, and the authors also submitted alignments for the mitochondrial and nuclear data.

The authors reported several field collecting permits issued by the Costa Rica, Panamá, and Nicaragua government offices and they seem appropriate.

Experimental design

The manuscript contains original primary research; the research question is well defined and relevant. It is also stated how this research fills a knowledge gap.

However, the methods used need some revision.

Basically, the study aims at obtaining robust phylogenetic and species delimitation hypotheses for the Craugastor podiciferus species group (Anura: Craugastoridae) of Isthmian Central America using mitochondrial and nuclear data, and at performing an ancestral area reconstruction for the group. The mitochondrial dataset is composed of fragments of two mitochondrial genes that have been used extensively in anuran systematics, and the nuclear data are restriction site-associated DNA sequence data (= RADseq data) from several thousand of loci.

The data collection was not ideal. The mitochondrial and RADseq data were obtained using standard methods and the taxonomic and geographic sampling was very good for the mitochondrial dataset; however, the sampling for the RADseq dataset was much less complete: some taxa were not included and the number of samples per taxon was considerably lower than in the mitochondrial dataset. Considering that at the time that the study was performed generating RADseq data was more expensive than it is now, this is understandable, but the the fact remains that the sampling is wanting. In the end, the authors recognize 23 distinct species, and the RADseq dataset included 42 samples; that is, barely two samples per species in average, although in fact some of the lineages were not represented in the dataset.

The data collection methods are rather standard; they are well described, only in too much detail. Their description should be shortened significantly, perhaps to one-half of its current extent.

I am also concerned about some aspects of the bioinformatics processing of the RADseq data. The processing of this data in the pipeline ipyrad is generally correct; however, it is also described in too much detail, and it was superficial in some aspects. First, the authors use a single method to estimate the optimum clustering threshold, but several others methods for this purpose have been proposed recently by McCartney and Melstad (2019) (https://doi.org/10.5281/zenodo.2540263), and it is advisable to use at least some of them for a corroboration of the threshold. Second, although the authors tried to find appropriate settings for several parameters such as the clustering threshold and the proportions of ambiguous (N) sites, indels, SNPs, etc. in the generation of assemblies, they did not do the same for the taxon coverage (i.e., the minimum number of samples with data for a given locus to be included in the final assembly). This parameter determines the number of retrieved loci and the proportion of missing data in the resulting assembly, which in turn may affect the topology and support of the inferred phylogenetic hypotheses. See for instance Hovmöller et al. (2013) (https://doi.org/10.1016/j.ympev.2013.06.004), Xi et al. (2016) (https://doi.org/10.1093/molbev/msv266), and Crotti et al. (2019) (https://doi.org/10.1111/zsc.12335), among others.

Also, the authors performed several analyses of the RADseq concatenated data, but no analysis to infer a species tree. Given that the study group is composed of closely related lineages, the presence of incomplete lineage sorting is a possible source of conflict and methods to infer species trees are designed to minimize this problem (e.g., SVDquartets, Astral, etc.)

I am also concerned about the species delimitation (=SD) methods used. The authors performed SD analyses separately with the mitochondrial and nuclear datasets using four SD methods (GMYC, PTP, mPTP, and BPP) with both datasets and two more (ABGD and GD) with the first. This alone would have resulted in many SD hypotheses; however, the number of hypotheses was escalated by performing additional analyses with several methods.

For instance, for PTP, a method that requires an input tree, they performed three analyses of the mitochondrial dataset, using as input trees each of the three phylogenetic hypotheses for this dataset (generated by RAxML, MrBayes, and Beast). Similarly, they performed two analyses of the RADseq dataset, using as input trees each of the two phylogenetic hypotheses for this dataset (generated by RAxML and Beast, and MrBayes). However, the SD analyses of the RADseq dataset were also performed independently for each of the three main clades in the group. Thus, they performed two additional SD analyses of the major clades in the group using as input trees each of the above phylogenetic hypotheses.
Similar multiplication of analyses happened with other methods. This resulted in a large number of SD hypotheses, and considerable discordance among them, as it is recognized by the authors (lines 435—449). Thus, interpretation of the results is not straightforward, and the authors came up with criteria to recognize species or propose “unconfirmed candidate species” that seem convenient but may not be supported by all the SD methods used (e.g., “…unnamed species can be suggested if they are supported by all the combinations of methods within a dataset [mtDNA or nuDNA]” and “additional unnamed species can be suggested to reconcile discordant results among mtDNA and nuDNA to avoid synonymizing named species that were supported by a dataset [mtDNA or nuDNA]; lines 353--357).” They also use a SD criterion that was not mentioned: “additional unnamed species can be suggested if morphological evidence distinctively supports monophyletic clades found in both mtDNA and nuDNA” (lines 357—358) (Note that all clades are monophyletic).

Just as different methods of phylogenetic analysis (distance methods, parsimony, maximum likelihood, etc.) make different assumptions, use different models, so do SD methods. So, only the most appropriate methods for the datasets at hand should be used. According to the authors, using the mitochondrial data some methods identified > 60 candidate species, while others identified only 13. Similarly, using the nuclear dataset some methods identified a single canidate species for the entire complex, whereas others identified up to 36. Clearly, some of these results are not credible.

When data from thousands of nuclear loci are available, SD based on data from two mitochondrial genes, and furthermore mainly on genetic distances, seems unconvincing. Given that the study group is composed of closely related species, the RADseq dataset could provide enough resolution and much more statistical power for both phylogenetic inference and species delimitation than the mitochondrial dataset. In addition, mitochondrial introgression is common, often distorting mitochondrial phylogenies, and phylogenetic analysis based on nuclear data can minimize that problem. In fact, mitochondrial introgression is probably the main reason behind the discrepancies between the mitochondrial and nuclear phylogenies. Thus, I would suggest that tree-based SD methods should best used with the nuclear dataset only; however, unfortunately the poor taxon and geographic sampling of the RADseq dataset may affect the resolution of these methods.

The authors used as another SD method automatic barcode gap discovery or ABGD. This method has been used to quickly identify candidate species but that should be complemented with other evidence, since rates of substitution may vary among lineages, as can levels of variation within and between species, which makes it difficult to assume a universal threshold. For their last “SD approach” the authors used just simple genetic distances based on each mtDNA dataset separately, which as the authors recognize is not a formal, rigorous SD method and has the same problems as ABGD. This criterion was actually the most used in this manuscript.

The last SD method used by the authors was BPP. Although this method seems solid and it is widely used for SD, there has been a recent concern that it tends to recognize more species than actually exist. A discussion of this problem can be found in Leaché et al. (2019) (https://doi.org/10.1093/sysbio/syy051), who proposed the use of BPP to calculate a genealogical divergence index for a more stringent SD. I suggest the authors consider using this new approach. Another recent approach is the fixed differences analysis with the software dartR (https://cran.r-project.org/web/packages/dartR).

In summary, although the authors used a number of SD methods, there are some more recent, rigorous methods that can take advantage of their RADseq data, like the genealogical divergence index or fixed differences approaches just mentioned. Another approach, if not strictly a SD method, is the evaluation of population genetic structure with programs such as Structure, Geneland, or ConStruct, which can shed light on the number of genetically distinct populations in a group.

Validity of the findings

The manuscript makes a valuable contribution to the systematics and biogeography of the Craugastor podiciferus species group. The collected data are of great value, even though the taxonomic and geographic sampling for the RADseq data is deficient. Although the bioinformatics processing of the RADseq data could be improved, it is likely that the basic phylogenetic hypothesis obtained for the group would not change significantly. This hypothesis and the ancestral area reconstruction generated from it are interesting and will provide a strong stimulus for additional studies of the group.

However, the SD analyses should be improved. There were considerable differences between the results of the SD analyses performed on the mitochondrial and nuclear datasets and among results from the analyses with different methods with each dataset. This was to be expected and the interpretation of results is not straightforward. Because the sampling of the RADseq dataset was deficient, identification of several “unconfirmed candidate species” relies mostly or only on mitochondrial data, and more specifically on genetic distance data. Thus, whether or not they truly represent distinct species is uncertain. However, the treatment of these “unconfirmed candidate species” precisely as “unconfirmed” means that they need to be corroborated by additional data, more rigorous analyses, and evidence from other sources. That is, their species delimitation should be seen as a pioneering study that suggests the existence of considerable undescribed diversity; but this existence must be confirmed before it is described.

To be fair, it is likely that many or most of the named species and the “unconfirmed candidate species” recognized by the authors be actually supported as distinct species, but this needs to be corroborated with more rigorous analysis and additional data.

An especially problematic proposal is that of six “unconfirmed candidate species” in the C. podiciferus clade based on unpublished morphological and advertisement call data and a previous study by other workers. As the authors admit, these lineages were not supported by their own analyses. Then, the authors should present the data to support their proposal, or simply refrain from present it.

Additional comments

Additional corrections and comments are noted in the manuscript file (attached)

The manuscript has value and I believe it is interesting and appropriate for the journal. It presents robust phylogenetic and biogeographical hypotheses for a diverse group of poorly studied group of amphibians in poorly known region of Central America, and even if not all the named and “unconfirmed candidate species” proposed in the manuscript turn out to be distinct, the manuscript still reveals an astonishing biodiversity that deserves more study. However, I believe the species delimitation analysis and the English language in general should be revised.

Annotated reviews are not available for download in order to protect the identity of reviewers who chose to remain anonymous.

---

## Round 0.2 · accepted · Accept

The authors have addressed adequately the reviewers' suggestions. The manuscript is ready for publication.